# Further Evidence of Neuroprotective Effects of Recombinant Human Erythropoietin and Growth Hormone in Hypoxic Brain Injury in Neonatal Mice

**DOI:** 10.3390/ijms23158693

**Published:** 2022-08-04

**Authors:** Simon Klepper, Susan Jung, Lara Dittmann, Carol I. Geppert, Arnd Hartmann, Nicole Beier, Regina Trollmann

**Affiliations:** 1Division of Pediatric Neurology, Department of Pediatrics, Friedrich-Alexander Universität Erlangen-Nürnberg, Loschgestr. 15, 91054 Erlangen, Germany; 2Institute of Pathology, Friedrich-Alexander Universität Erlangen-Nürnberg, Krankenhausstr. 8, 91054 Erlangen, Germany

**Keywords:** angiogenesis, vasculogenesis, VEGF-A, angiopoietins, hypoxia, blood–brain barrier, occludin, tight junction proteins, hypoxic brain injury, neurovascular unit

## Abstract

Experimental in vivo data have recently shown complementary neuroprotective actions of rhEPO and growth hormone (rhGH) in a neonatal murine model of hypoxic brain injury. Here, we hypothesized that rhGH and rhEPO mediate stabilization of the blood–brain barrier (BBB) and regenerative vascular effects in hypoxic injury to the developing brain. Using an established model of neonatal hypoxia, neonatal mice (P7) were treated i.p. with rhGH (4000 µg/kg) or rhEPO (5000 IU/kg) 0/12/24 h after hypoxic exposure. After a regeneration period of 48 h or 7 d, cerebral mRNA expression of *Vegf-A*, its receptors and co-receptors, and selected tight junction proteins were determined using qRT-PCR and ELISA. Vessel structures were assessed by Pecam-1 and occludin (Ocln) IHC. While *Vegf-A* expression increased significantly with rhGH treatment (*p* < 0.01), expression of the *Vegfr* and TEK receptor tyrosine kinase (*Tie-2*) system remained unchanged. RhEPO increased *Vegf-A* (*p* < 0.05) and *Angpt-2* (*p* < 0.05) expression. While hypoxia reduced the mean vessel area in the parietal cortex compared to controls (*p* < 0.05), rhGH and rhEPO prevented this reduction after 48 h of regeneration. Hypoxia significantly reduced the Ocln^+^ fraction of cortical vascular endothelial cells. Ocln signal intensity increased in the cortex in response to rhGH (*p* < 0.05) and in the cortex and hippocampus in response to rhEPO (*p* < 0.05). Our data indicate that rhGH and rhEPO have protective effects on hypoxia-induced BBB disruption and regenerative vascular effects during the post-hypoxic period in the developing brain.

## 1. Introduction

Hypoxia and ischemia (HI) remain the most common causes of acquired brain injury in term and preterm infants, leading to acute encephalopathy, high mortality, long-term sensorimotor, cognitive, and behavioral disorders, and refractory epilepsy [1]. Specific pharmacological therapies to prevent cerebral lesions and developmental disorders in at-risk preterm and term infants at risk are not available. However, experimental studies have provided evidence of the protective effects of exogenous neurotrophic growth factors such as insulin-like growth factor 1 (IGF-1), recombinant human growth hormone (rhGH), and recombinant human erythropoietin (rhEPO) in the treatment of HI brain injury during the perinatal period [2,3,4,5,6]. These novel therapies target the complex cytotoxic cascade of cerebral hypoxic injury during the acute and subacute HI period that involves oxidative stress responses, glutamate-induced excitotoxicity, pro-inflammatory pathways, and neurovascular damage [7,8]. Consequently, increased vulnerability of maturational processes such as neuronal and glial proliferation [9], angiogenesis [10,11], and blood–brain barrier (BBB) development [12] continues for days and weeks, resulting in apoptotic neuronal death in specific brain regions [11]. However, hypoxia activates a broad spectrum of endogenous cytoprotective mechanisms that are primarily regulated by specific target genes of hypoxia-inducible transcription factors (HIFs; [13]), which are involved in crucial cell metabolism and survival mechanisms, but also in regenerative glial and vasoactive processes and protection of the developing BBB [14,15].

Angiogenesis and neovasculogenesis represent crucial mechanisms in post-HI regeneration in developing brain injury that are regulated by a broad range of vasoactive growth factors that are partially under the control of HIFs such as vascular endothelial growth factor (VEGF), VEGF receptors 1 and 2 (VEGFR-1/2), angiopoietins 1 and 2 (ANGPT-1/2; [14]) and TEK receptor tyrosine kinase (TIE-2). In the developing brain, VEGFR-2, ANGPT-1, and TIE-2 are expressed predominantly in endothelial cells and endothelium-associated pericytes and are regulated by HIF-2. However, cell-type-specific molecular changes in the subunits of the neurovascular unit in response to hypoxic injury are not fully understood [9,16].

The crucial role and specific properties of vascular endothelial cells and interendothelial tight (TJ) and adherens (AJ) junction proteins in the functional maintenance of complex intercellular interactions between the specific neurovascular unit cell types and BBB integrity have been characterized in the developing rodent brain [17]. Rapid disruption of the developing BBB, composed of neurons, glial cells, and microvessels, in HI brain injury results in increased paracellular secretion of pro-inflammatory proteins, diffusion of various neurotoxic substances, and diffuse cerebral edema, contributing to HI neuronal degeneration and apoptosis [12]. While degradation of specific TJ proteins is believed to be the main mechanism leading to disruption of the developing BBB in response to hypoxia [12,17], the overall process remains unclear. For example, the exact role and regulation of various TJ proteins such as claudin-1 (CLDN1), TJ protein 1 (ZO-1), and occludin (OCLN), and adherens proteins such as catenins and cadherins, which are essential for the specific barrier functions during early brain development, remain to be determined [12,18].

Erythropoietin (EPO) is a hematopoietic growth factor regulated by HIF-2 and an anti-inflammatory, anti-apoptotic, and pro-survival factor upon cerebral hypoxia [19] with neuroprotective potential in HI brain injury in neonatal rodents [4,5,20]. These protective effects are mediated via its binding to EPO receptor (EPOR) homodimers and Janus kinase 2 (Jak2) phosphorylation, multiple signaling pathways, including phosphoinositide 3-kinase (PI3K)/protein kinase B (Akt), mitogen-activated protein kinase (MAPK)/extracellular signal-regulated kinase 1/2 (ERK-1/2), signal transducer and activator of transcription 5 (Stat5), and nuclear factor κ-light-chain-enhancer of activated B cells (NF-κB), and modification of the transcriptional activity of anti-apoptotic factors [19,20].

Beyond acute effects, rhEPO enhances longer-term regenerative cerebral processes such as angiogenesis via the PI3K/Akt signaling pathway [3,19] associated with upregulation of VEGF and ANGPTs and increased vessel density in the white matter, hippocampus, and parietal cortex [5,21,22]. Moreover, rhEPO increases cerebral protein levels of the TJ protein OCLN in neonatal rodent stroke [23], ventilation-induced cerebral white matter injury in preterm lambs [24], and traumatic brain injury [25], indicating its protective potential against hypoxia-induced BBB disruption. However, studies on the longer-term regenerative effects of rhEPO on neurovascular repair and protection of the BBB during the early stages of brain maturation are limited.

Among endogenous neurotrophic pathways, the growth hormone (GH)/IGF-1 axis is crucially involved in early brain development [26,27] and regenerative cellular mechanisms in HI injury to the developing brain [28]. These protective effects are mediated via GH receptor (GHR) binding, which is primarily expressed in the hippocampal dentate gyrus (DG) and subventricular zone (SVZ), modulating neurogenesis, synaptogenesis, and learning and memory functions [29]. RhGH reduced cerebral apoptosis and pro-inflammatory mechanisms in HI brain injury in neonatal rodents [6,7,27]. Furthermore, the GH/IGF-1 axis was also crucially involved in post-hypoxic neovasculogenesis in ischemic adult rat myocardium and ischemic mouse retina [30,31] and in protection of the developing BBB against HI-induced disruption [32]. However, the underlying mechanisms and longer-term effects of rhGH on post-hypoxic angiogenesis and BBB integrity in the developing brain remain to be determined.

This study characterizes hypoxia-induced vascular changes in hypoxic developing brain injury of 7-day-old mice and assesses longer-term neurovascular effects of rhGH and rhEPO, which are both well-characterized neurotrophic growth factors, with a particular focus on vascular development and the regulation of oxygen-dependent vasoactive factors and TJ proteins. We examine cerebral regulation of VEGF and its receptors and co-receptors, vascular structural changes, and expression of specific BBB-stabilizing TJ proteins in hypoxic brains of 7-day-old mice treated with rhEPO or rhGH in two different regenerative intervals (48 h, 7 d).

## 2. Results

### 2.1. Differential Changes of Angiogenesis-Associated Gene Expression and Vascular Structures in Response to rhGH in the Hypoxic Developing Mouse Brain

For comparison of longer-term effects of exogenous growth factors on neurovascular development, we first analyzed the degree of hypoxia-induced disturbances of angiogenesis-associated gene expression and vessel development during the early stage of brain maturation. Cerebral expression of oxygen-sensitive genes involved in angiogenesis and neovasculogenesis was assessed in brain lysates of neonatal mice exposed to acute systemic hypoxia at P7, followed by different regeneration intervals. In addition, we quantitatively explored developing vessel structures by endothelial PECAM-1 staining in coronal brain sections to gain additional insights into structural vessel development after hypoxic brain injury and its changes in response to neurotrophic growth factor treatments.

There were no significant differences in cerebral expression of the oxygen-dependent, HIF-1-regulated vasoactive gene VEGF-A (Figure 1A,B) and its receptors (VEGFR-1/2; Appendix A) between hypoxia-exposed brains and normoxic controls during subacute (48 h) and chronic (7 d) post-hypoxic periods. Considering the crucial role of VEGF co-receptors in vessel development and vessel stabilization, we determined the cerebral expression of *Angpt-1/2* and *Tie-2*, which are also partially regulated by HIF-1. Similarly, we did not observe significant changes in the cerebral expression of *Angpt-1/2* and *Tie-2* (Figure 1E,F and Appendix A) in hypoxia-exposed brains compared to controls beyond the acute post-hypoxic period. However, while acute hypoxia led to a significantly lower mean vessel area in the parietal cortex (Figure 2A,B; *p* < 0.05) compared to controls, the observed difference in mean vessel area between normoxic and hypoxic hippocampal regions did not reach statistical significance (Figure 2C,D; Appendix A).

To examine the potential effects of rhGH on cerebral vascular–endothelial structures in the subacute and chronic post-hypoxic phase (48 h, 7 d), we administered rhGH at the end of hypoxia exposure (0 h) followed by repeated injections 12 and 24 h later and assessed the effects at 24 h and 6 d intervals post-injection. We observed significantly higher VEGF-A expression in the brain 7 d after hypoxia exposure in response to this high-dose treatment (Figure 1B; *p* < 0.05) and VEGF-A protein levels in normoxic and hypoxia-exposed brains after a regeneration interval of 48 h compared to controls (Figure 1C; *p* < 0.01). This pattern was associated with unchanged mean mRNA levels for *Angpt-2* (Figure 1E,F), *Vegfr-1, Vegfr-2* (Appendix A), *Angpt-1*, and *Tie-2* compared to controls (Appendix A). IHC analysis also showed rhGH treatment transiently increased mean vessel area in the hypoxia-exposed parietal cortex (Figure 2A; not significant (n.s.)) and consequently attenuated the hypoxia-induced decrease in vessel area without markedly different branching of hippocampal and cortical vessels in comparison to VT controls (Appendix A).

### 2.2. Differential Changes of Angiogenesis-Associated Gene Expression and Vascular Structures in Response to rhEPO in the Hypoxic Developing Mouse Brain 

Considering the well-characterized regenerative effects of rhEPO, we further assessed the expression of angiogenesis-associated genes in response to high-dose rhEPO in developing mouse brains during the subacute and longer-term post-hypoxic periods. We found a transient increase in VEGF-A protein levels (Figure 1C,D) and significantly increased *Angpt-2* expression (Figure 1E,F) but unchanged *Angpt-1* and *Tie-2* expression (Appendix A) in response to rhEPO in hypoxic brains compared to controls. There was also a higher mean vessel area in the hippocampus and, to a lesser extent, in the parietal cortex of hypoxic pups treated with rhEPO compared to controls, indicating partial attenuation of a hypoxia-induced decrease in vessel area (Figure 2A). Again, this effect was transiently observed during the subacute post-hypoxic period, and there were no marked changes in vessel branching compared to controls (Appendix A).

### 2.3. Effects of rhGH on Hypoxia-Induced Disruption of Developing BBB 

We next explored the potential therapeutic effects of neurotrophic growth factors on developing BBB structures with hypoxic injury to the developing mouse brain during the subacute and chronic regenerative periods. Therefore, we analyzed gene and protein expression of specific TJ proteins, which are highly vulnerable to hypoxia, in vascular endothelial cells from the hypoxic mouse brain to determine whether rhGH and rhEPO attenuate the detrimental changes induced by hypoxic injury.

We found slightly decreased *Ocln* expression (Figure 3B), but not *ZO-1*, *CLDN1*, and claudin 5 (*CLDN5*) expression (Appendix A), in response to hypoxia in whole brain lysates compared to controls after a 7 d regeneration period. Therefore, we next assessed OCLN protein levels with IHC, finding fewer OCLN^+^ cells in the hypoxic than in the normoxic parietal cortex (Figure 4A,B and Figure 5), indicating disruption of endothelial TJ protein structures as a consequence of acute global hypoxia. Notably, the mean OCLN^+^ area significantly increased in the hypoxic parietal cortex of pups treated with rhGH compared to controls, particularly after 48 h of reoxygenation (Figure 4A). However, the larger OCLN^+^ area observed in the hippocampal areas in the rhGH treatment group did not reach statistical significance (Figure 4C,D). Representative photomicrographs of endothelial co-stainings with OCLN and PECAM-1 of hypoxia-exposed and rhGH-treated brains compared to controls at the level of the parietal cortex are shown in Figure 5.

### 2.4. Effects of rhEPO on Hypoxia-Induced Disruption of Developing BBB

To assess the potential effects of rhEPO on vascular–endothelial structures of the developing BBB, we assessed gene and protein expression of specific endothelial TJ proteins susceptible to hypoxic injury. We found significantly increased *OCLN* expression (Figure 3; *p* < 0.05), but not *ZO-1*, *CLDN1*, and *CLDN5* expression (Appendix A), in response to rhEPO compared to normoxic controls. In addition, we detected significantly higher OCLN protein levels in both hypoxic cortical and hippocampal vessels in response to rhEPO compared to controls during the subacute post-hypoxic period (Figure 4; *p* < 0.05). After a 7 d regeneration period, OCLN protein levels did not differ between normoxic and hypoxic brains, suggesting sufficient compensatory effects of the transient increase in OCLN expression during the early post-hypoxic period. Representative examples of co-stainings with OCLN and PECAM-1 of hypoxia-exposed and rhEPO-treated brains in comparison to controls at the level of the parietal cortex are shown in Figure 5.

### 2.5. Anti-Inflammatory Effects of rhGH and rhEPO Attenuate Hypoxia-Induced Disruption of the Developing BBB 

Based on the well-characterized interactions between rhEPO and inflammatory cytokine expression, we hypothesized that the anti-inflammatory effects of rhEPO may modify hypoxia-induced disruption of the developing BBB. Therefore, we assessed specific chemokines known to be involved in the hypoxic stress response in the developing mouse brain, including IL-2, IL-6, and IL-10, after a 48 h regeneration period. We found significantly lower IL-2 protein levels (Figure 6A) and significantly higher IL-10 protein levels (Figure 6C) in rhEPO-treated brains compared to controls. We found similar results with brain tissues treated with rhGH, with significantly lower IL-2 protein levels (Figure 6A) and significantly higher IL-10 protein levels (Figure 6C) in rhGH-treated brains compared to controls.

## 3. Discussion

To gain more specific insights into potential therapeutic targets in hypoxia-induced cerebral angiogenesis and disturbances of BBB integrity in hypoxic developmental brain injury without experimental ischemia, we assessed the hypothesis that neurotrophic growth factors rhGH and rhEPO mediate longer-term regenerative effects on hypoxia-induced vascular–endothelial disturbances and BBB disruption in hypoxic brain injury during the early stage of brain maturation. Our group has recently characterized the anti-apoptotic and anti-inflammatory properties of rhEPO and rhGH in the established neonatal murine model of acute hypoxic brain injury in the same experimental setting used here [5,32]. This study focused on the subacute and longer-term post-hypoxic period and found that rhEPO and rhGH treatment increased cerebral angiogenesis, assessed by mean vessel area, induced a significant accumulation of the vasoactive VEGF-A protein, and protected BBB integrity, as indicated by prevention of OCLN degradation after hypoxic injury to the neonatal mouse brain. In addition, the observed effects of rhEPO on the Angpt-2/Tie-2 system confirm its vascular regenerative properties. Therefore, our results demonstrate that both rhEPO and rhGH mediate regenerative effects on vascular–endothelial structures in hypoxic developmental brain injury and might attenuate hypoxia-induced injury of the neurovascular unit.

Our finding that the expression of the vasoactive and neurotrophic factor VEGF-A was transiently increased after acute hypoxia is consistent with previous observations by our group [5] and others [33] and represents a characteristic compensatory response of the developing brain to hypoxia [3]. However, the dual functions of VEGF-A in the acute period of cerebral hypoxia prevent a coherent estimation of this response to HI and depend on the degree of hypoxia and time [34,35]. Upregulation of VEGF-A during the immediate acute period of hypoxia may mediate increased vascular–endothelial and BBB permeability in adult and neonatal HI brain injury [34,36], predisposing the developing brain to edema. However, VEGF-A upregulation protected hypoxic neurons from cell death in adult and neonatal HI brain injury [37], and rhVEGF-165 prevented HI-induced decrease in cerebral vascular density in newborn piglets [10]. HI-induced cerebral VEGF accumulation mediated by HIF-1 [38] binding to VEGFR-2, activating several downstream signaling pathways in endothelial cells, pericytes, microglia, and oligodendrocytes [39,40] of the neurovascular unit, such as the PI3K/Akt [41] and NF-kB [42,43] pathways involved in a dynamic modification of post-hypoxic angiogenesis. However, data on the regulatory mechanisms of regenerative angiogenesis in experimental global brain hypoxia are limited.

Consistent with our findings, the HIF-dependent VEGF/VEGFR stress response and the increased BBB permeability it causes have been shown to return to physiological levels within 48–72 h post-HI brain injury [12,44]. In this study, we found that cerebral VEGF-*A* gene and protein expression and the gene expression of its receptors (*Vegfr-1/2*) and co-receptors (*Angpt-1/2* and *Tie-2* system) did not differ significantly between hypoxia-exposed brains and controls after regeneration periods of 48 h and 7 d in neonatal mice. In contrast, our brain-region-specific IHC analysis of vascular–endothelial effects of hypoxia found a decreased mean vessel area in the hypoxic parietal cortex and hippocampal areas of developing brains assessed during the subacute and longer-term periods of regeneration after acute hypoxia. These observations are consistent with previous findings that showed transient endothelial upregulation of VEGF in vessels of the SVZ and periventricular and subcortical white matter regions during a 24–48 h post-HI period in the fetal ovine brain [44] and in neonatal rodent stroke models [21]. In addition, lower vessel density was also observed in the hypoxic developing piglet brain after a reoxygenation period of 72 h [10]. Moreover, a slight increase in angiogenesis was found in a neonatal rodent stroke model after 48 h of regeneration [21] and in our mouse model of acute global brain hypoxia after 72 h of regeneration [5]. Therefore, different species, hypoxia models, regions of interest, and region-specific regulation may explain variable observations on cerebral VEGF-A regulation and angiogenesis during the post-hypoxic repair period.

### 3.1. RhGH Has Pro-Angiogenic Effects in the Hypoxic Developing Mouse Brain 

A broad spectrum of neuroprotective rhGH effects have been characterized in vitro and in vivo, where autocrine and paracrine rhGH effects are mediated either by neuronal transmembrane GHR binding or activation of neuronal and glial IGF-1 signaling [27], particularly in brain regions with highly active neurogenesis such as hippocampal DG and SVZ [26,27]. In addition to anti-apoptotic actions on neurons and glial cells shown in hippocampal cell cultures exposed to oxygen and glucose deprivation injury [45], during brain regeneration after HI injury in the developing mouse brain [6,27,46] and adult rodent stroke models [47], the oxygen-sensitive GH/IGF-1 system promotes synaptogenesis, dendritic growth, neurotrophic factor expression, including brain-derived neurotrophic factor (BDNF), IGF-1, EPO, and VEGF [27,48]. Moreover, stimulation of angiogenic factors [27] and angiogenesis in the perifocal area and penumbra in experimental stroke models have been reported [48]. Our data suggest potential pro-angiogenic and regenerative rhGH effects in the hypoxic developing mouse brain. Focusing on the cerebral regulation of vasoactive growth factors during the subacute post-hypoxic period, we showed an increase in VEGF-A levels in rhGH-treated mouse brains and an increase in mean vessel area in the parietal cortex of hypoxia-exposed mice. These observations indicate that rhGH restored the hypoxia-induced decrease in angiogenesis after 7 days of regeneration in our experimental setting. Similarly, Li et al. [27] found a significant upregulation of the neurotrophic and angiogenic factors VEGF, EPO, and IGF-1 in response to 50 µg/kg rhGH in the developing rat brain after chronic intermittent hypoxia and a 7-day recovery period.

Regulatory signaling pathways are poorly understood. As synergistic cytoprotective effects of the cerebral GH/IGF-1 system and hypoxia-sensitive growth factors have been suggested, we hypothesize that vascular regenerative effects might result from rhGH-induced upregulation of endogenous EPO and downstream activation of ERK1/2 signaling [32]. Downstream effects of the GH/IGF-1 system are mediated by the PI3K/Akt, Akt/mammalian target of rapamycin (mTOR), and c-Jun N-terminal kinase (JNK) pathways in neuronal cells [49]. The oxygen-sensitive growth factor IGF-1, which acts GH-dependently and -independently in the developing brain, activates paracrine PI3K/Akt and MAPK signaling by binding to neuronal GHRs [50]. Therefore, we hypothesize that the vascular regenerative effects of rhGH are mediated via activation of PI3K/Akt signaling, enhancing the production of endogenous VEGF and inducing post-hypoxic vascular–endothelial survival and angiogenesis via VEGF/VEGFR-2 binding [41]. However, region-, cell-type-, and age-specific rhGH effects on the neurovascular unit have to be considered and remain to be clarified in detail. Altogether, rhGH treatment might create the opportunity to combine regeneration of the neurovascular unit by activation of the endogenous vascular–regenerative growth factors VEGF, EPO, and IGF-1 and neuroprotection [48,51] in hypoxic developing brain injury.

### 3.2. RhEPO Has Pro-Angiogenic Effects in the Hypoxic Developing Mouse Brain

Previous experimental studies using rodent models of HI brain injury have shown short-term neuroprotective rhEPO effects, including anti-inflammatory, anti-oxidative, and anti-apoptotic actions and protection of developing neurons, astrocytes, and oligodendroglia from hypoxia-induced apoptosis [19,52]. Moreover, its neuroregenerative activities in stimulating neurogenesis [8,53] and angiogenesis [8,22,54] via activation of PI3K/Akt and ERK1/2 [21] are increasingly gaining attention in adult and neonatal rodent models of HI brain injury. Beyond short-term regeneration periods, regenerative pro-angiogenic effects of rhEPO have been shown in rodent cortex and hippocampus up to 90 and 35 days after traumatic and HI brain injury, respectively [55,56]. In agreement with our findings here, rhEPO treatment induced angiogenesis via VEGF production in neural progenitor cells and VEGFR-2 activation in cerebral endothelial cells in an in vitro co-culture system of mouse brain endothelial cells (MBECs) and neural progenitor cells [54]. Using a neonatal HI rat model, rhEPO (1000–2000 IU/kg i.p. at day 0, 2, 4, and 6 after HI) significantly increased the number of functional vessels in both perilesional and distant brain areas compared with vehicle-infused HI animals, while functional vessels were dramatically decreased in the perilesional area one day after HI and remained below age-matched controls over the 14 days of the study [8]. This finding is consistent with our results, although we used a neonatal mouse model of global hypoxia without experimental ischemia and a higher rhEPO dose (5000 IU/kg i.p.). RhEPO treatment led to transient activation of VEGF gene and protein expression and VEGFR-2 gene expression and the simultaneous stimulation of regionally accentuated angiogenesis, indicating functional stabilization of vessel structures. Furthermore, this is consistent with previous studies that found rhEPO/heterodimeric EPOR binding results in a region-, time-, and cell-type-specific regulation of VEGF and ANGPT-2 expression in HI brain injury in neonatal rodents [21], while the pro-angiogenic effects of ANGPT-2, such as enhanced capillary endothelial cell proliferation and stimulating endothelial cell sprouting and migration and microvessel density depends on the presence of VEGF and Tie-2 binding [43,57]. Previous studies by our group [5] and others [21,22,40] underscore the crucial role of HIF and PI3K/Akt pathways in the regulation of neurovascular remodeling after hypoxic neonatal brain injury and that rhEPO effects and EPOR regulation differ based on the degree of injury and treatment regimens.

### 3.3. Prevention of BBB Damage in the Hypoxic Developing Mouse Brain 

Degradation of specific TJ proteins is believed to be the primary mechanism of hypoxia-induced disruption of the developing BBB, which is partially under the control of HIFs and associated with detrimental consequences for neuronal survival [12,17,18]. In this study, we showed that exposure of 7-day-old mice to acute systemic hypoxia mildly decreased endothelial OCLN protein expression, but not OCLN, ZO-1, CLDN1 and CLDN5 mRNA levels, in the developing brain assessed upon a regeneration period of 48 h and 7 days, indicating a high sensitivity of *Ocln* expression to acute hypoxia. Using a neonatal murine stroke model, Fang et al. [18] observed reduced *Ocln* expression, but also *Cldn5*, p120/Catenin (*Ctnnd1*), VE-Cadherin (*Cdh5*), and β-Catenin (*Ctnnb1*) in cortical and hippocampal regions. In addition, Ma et al. [12] found an increased vulnerability of the BBB associated with downregulation of *Ocln* and *Ctnnb1* in a similar HI experimental setting. OCLN degradation was suggested to be caused by HI-induced upregulation of metalloproteinases (MMPs) such as MMP-2 and MMP-9, tumor necrosis factor α (TNF-α), interleukin 1β (IL-1ß), MAPK, protein kinase C (PKC), ERK1/2, NF-κB, and HIFs [35,58]. In addition, in vitro studies on retinal endothelial cells showed OCLN activation phosphorylation and TJ permeability by VEGF [59].

Protective effects of endogenous neurotrophic factors on the time-dependent, biphasic pattern of developing BBB leakage after HI brain injury [60] have been assumed, but published data are limited [21,54]. Our findings show beneficial effects of both exogenous neurotrophic treatments rhEPO and rhGH on the maintenance of the TJ protein OCLN in vascular endothelial cells in hypoxic developing brain injury, suggesting stabilizing effects on BBB integrity by both treatments in hypoxic developing brain injury. The temporary increase in OCLN^+^ endothelial cells occurred in a region-specific manner with a higher OCLN expression primarily in the parietal cortex with rhGH treatment and in the parietal cortex and dorsal hippocampus with rhEPO treatment. Concerning the pathways mediating the protective rhEPO effects on OCLN expression and BBB integrity, inhibition of MMP-2 and MMP-9 [23] and decreased TNF-α levels and NF-*κ*B activation [42] have been reported. The decrease in IL-2 and increase in IL-10 protein levels in whole brain lysates in response to rhEPO shown here underscore the possible regulatory significance of anti-inflammatory rhEPO effects on hypoxia-induced OCLN degradation. Both, IL-2 and IL-10 are likely expressed by neuronal cells, astrocytes, and microglia [61,62,63]. While HI-triggered secretion of IL-2 by glial cells contributes to neuronal cell death [64] and disruption of the BBB [65], IL-10 is considered to be one of the crucial immunosuppressive, neuroprotective cytokines expressed within the brain [61]. In vitro and in vivo studies have shown that IL-10 dampens inflammatory responses [66] and prevents vascular dysfunction due to endothelial cell damage [67]. However, the hypothesis that anti-inflammatory rhEPO effects are of regulatory significance needs to be explored further, including extended analyses of TJ protein regulation in response to rhEPO at different time points after an acute hypoxic injury during early development. 

The regulatory effects of rhGH and the GH/IGF-1 system on TJ proteins of the neurovascular unit remain largely unknown. In this study, we showed that rhGH might stabilize OCLN expression during early development, indicating protective actions on the endothelial functions of cerebral microvessels. However, this protective effect may be the result of the activation of endogenous EPO in response to rhGH treatment [6,27], in which biological activity and neural and endothelial cellular responses of the GH/IGF-1 system are modified by several neurotrophic factors others than EPO, such as BDNF, fibroblast-growth factors, prolactin and its receptor, and neurotrophin 3 [68,69]. Nevertheless, we have shown that the anti-inflammatory properties of rhGH cause a significant decrease in IL-2 and increase in IL-10 levels, which together with the activation of ERK1/2 signaling, found to mediate its neuroprotective effects in an experimental rodent stroke model by suppressing oxidative stress and pro-apoptotic pathways [27,70], may contribute to protecting BBB integrity in response to rhGH treatment during early development. In the context of the functional duality of ERK1/2 in HI brain injury [71], further studies on cell type-specific and functional effects are ongoing. Moreover, future analysis of regulatory signaling pathways at early post-hypoxic time intervals is required.

In conclusion, our findings suggest that rhGH and rhEPO may stabilize post-hypoxic angiogenesis via the upregulation of vasoactive growth factors during early brain development. Moreover, they strongly support the hypothesis that both rhGH and rhEPO significantly contribute to preventing the degradation of TJ protein OCLN and hypoxia-induced BBB damage. Therefore, recombinant neurotrophic growth factors represent promising neuroprotective options in HI injury of the developing brain due to their regenerative activities in the neurovascular unit associated with neuronal protection during early development.

## 4. Materials and Methods

### 4.1. Animal Experiments

Animal experiments were performed according to protocols approved by the National Care Committee (Regierung Unterfranken; Wuerzburg, Germany) along with national and European laws on the protection of animals. Animals were provided with food and water ad libitum and housed with a 12 h light/12 h dark cycle. Pups were housed together with their dam to provide normal temperature and nutrition. A total of 128 C57BL/6NCrl neonatal mice (Charles River Laboratories; Sulzfeld, Germany) were block randomized at postnatal day 7 (P7) into the four treatment groups: (i) non-treated (NT; hypoxia, n = 16; normoxia, n = 16), (ii) vehicle-treated (VT; hypoxia, n = 16; normoxia, n = 16), (iii) rhGH-treated group (hypoxia, n = 16, normoxia, n = 16), and (iv) rhEPO-treated group (hypoxia, n = 16; normoxia, n = 16). After a regeneration period of 48 h (P9) or 7 d (P14), brains were dissected, snap-frozen in liquid nitrogen, and stored at −80 °C until required. The brains of a subgroup of pups (n = 3 per group) were perfused with Dulbecco’s phosphate-buffered saline (DPBS; pH 7.4) without Ca^2+^/Mg^2+^ (PAN-Biotech; Aidenbach, Germany) and fixed with 4% (*w*/*v*) paraformaldehyde (PFA; Carl Roth; Karlsruhe, Germany) for immunohistochemistry (IHC). Dissected brains were embedded in paraffin (Merck Millipore; Darmstadt, Germany). Appendix A presents the experimental design of this study.

### 4.2. Exposure to Acute Hypoxia

Using the established neonatal mouse model of acute systemic hypoxia [5], neonatal C57BL/6NCrl mice were exposed to acute systemic hypoxia (fraction of inspired oxygen (FiO_2_) 8%, 6 h) at P7 using an INVIVO_2_400 hypoxia workstation (Ruskinn Life Sciences; Bridgend, UK). Briefly, adaptation to the hypoxic environment was enabled by decreasing the oxygen concentration in steps of 2% O_2_ every 10 min. Age-matched controls were exposed to room air (FiO_2_ 21%) and otherwise held under similar conditions. Thereafter, pups were housed together with their dams in room air for regeneration periods of 48 h or 7 d. The degree of cerebral apoptosis induced by our experimental setting has been described previously [5].

### 4.3. Drug Treatment Regimens

RhGH (Genotropin MiniQuick; Pfizer; Berlin, Germany) was dissolved according to the manufacturer’s protocol as described by Jung et al., [6] and administered intraperitoneally (i.p.) at a dose of 4000 µg/kg body weight (total injection volume 10 mL/kg body weight) at 0, 12, and 24 h after hypoxic exposure. Age-matched VT controls were given 0.9% saline (NaCl) i.p. (total injection volume 10 µL/g body weight) or remained NT controls. The treatment regimen was performed according to our previous studies in 7-day-old mice [6] and reported pharmacokinetic studies in neonatal mice [27]. We assessed safety and efficacy in the range of 1000–4000 µg/kg body weight in our previous in vivo study, where a high-dose treatment did not affect hematocrit and hemoglobin levels or body weight development [6]. Drug toxicity was excluded in vitro by our previous studies on primary mouse cortical neurons (E14 and DIV6; Jung et al., 2021) and comprehensive observations from literature, including the use of supraphysiological doses of rhGH [27,51].

RhEPO (NeoRecormon; Roche; Grenzach, Germany) was diluted in 0.9% saline (B. Braun; Melsungen, Germany) to a final concentration of 1000 IU/mL and was given i.p. at a dose of 5000 IU/kg body weight (total injection volume 10 µL/g body weight) at 0, 12, and 24 h after hypoxic exposure. The treatment regimen was performed according to our previous studies in 7-day-old neonatal mice [5,72], consistent with previous pharmacokinetics [73] and high-dose rhEPO [14,22,56] studies in neonatal rodents. Age-matched VT controls were given 0.9% saline i.p. (total injection volume 10 µL/g body weight) or remained NT controls. There were no significant effects on body weight and rheological parameters, as previously described [5].

Significant anti-apoptotic effects of rhEPO [5] and rhGH [6] under the state of neonatal acute systemic hypoxia used here have previously been characterized and reported by our group.

### 4.4. Real-Time PCR

The expression of *Vegf-A*, its receptors *Vegfr-1/2*, its co-receptors *Angpt-1/2*, and *Tie-2* was determined by real-time quantitative reverse transcription (RT) polymerase chain reaction (qRT-PCR) as previously described [5]. Total RNA was extracted by TRIzol^TM^ according to the manufacturer’s protocol (Thermo Fisher Scientific; Schwerte, Germany). Genomic DNA was removed by DNase I treatment prior to complement DNA (cDNA) synthesis. Briefly, 1 μg of RNA was inoculated with 1 U of RNase-free DNase I (Promega; Mannheim, Germany) in a final volume of 10 µL of 1× reaction buffer at room temperature for 15 min. DNase I was inactivated by the addition of ethylenediaminetetraacetic acid (EDTA) and incubation at 65 °C for 15 min. RT was performed with diethylpyrocarbonate (DEPC; Sigma-Aldrich; Taufkirchen, Germany)-treated water, dT_16_ oligonucleotides (Eurofins MWG Operon; Ebersberg, Germany), and dN_6_ random hexamer oligonucleotides (Roche Diagnostics; Mannheim, Germany) added to a final concentration of 12.50 and 18.75 ng/µL, respectively, before the reaction mixture was heated to 70 °C for 5 min and then chilled on ice. RT was performed at 37 °C for 1 h in a 1 × transcription buffer containing 0.8 U/µL of recombinant RNasin ribonuclease inhibitor (Promega; Mannheim, Germany), 8 U/µL of Moloney murine leukemia virus (M-MLV) reverse transcriptase (Promega; Mannheim, Germany), 0.48 mM of each dNTP (Thermo Fisher Scientific; Schwerte, Germany), and DEPC-treated water. The quality and concentration of cDNA were assessed by UV–Vis spectroscopy (NanoDrop 2000C; VWR; Erlangen, Germany), adjusted with LiChrosolv water (Sigma-Aldrich; Taufkirchen, Germany) to a final concentration of 200 ng/μL, and stored at −80 °C until required. qRT-PCR was performed using the qPCR Core kit according to the manufacturer’s protocol (Eurogentec; Seraing, Belgium). Briefly, 1 μg of cDNA was assayed in the presence of 0.3–0.9 μM forward primer, 0.3–0.9 μM reverse primer, 0.2 μM of a gene-specific dual-labeled fluorescent probe (Biomers; Ulm, Germany), 3.5–5.0 mM magnesium chloride (MgCl_2_), 0.2 mM of each dNTP including dUTP, 0.14% (*v*/*v*) 6-carboxy-X-rhodamine (ROX), 0.02 U/µL of DNA polymerase, and 0.01 U/µL of Uracil–DNA glycosylase (Eurogentec; Seraing, Belgium). qRT-PCR was performed using a CFX96 Touch Real-Time PCR System (Bio-Rad; Munich, Germany) with the following thermal profile: one cycle at 50 °C for 2 min, followed by 95 °C for 10 min; 50 cycles of 95 °C for 5 s, followed by 60 °C for 1 min. PCR efficiency ranged between 97.9% and 102.0%. Relative target gene expression was calculated in relation to β-actin (*Actb*) and porphobilinogen deaminase (PBGD/*Hmbs*) using a PCR efficiency corrected model [74]. The primers used are listed in Table 1. All samples were assayed in duplicate.

### 4.5. Protein Extraction and Quantification

Protein extraction was performed according to [75]. Supernatants were harvested and stored at −80 °C until required. Protein concentrations were quantified using the Pierce BCA Protein Assay Kit (Life Technologies; Darmstadt, Germany) according to the manufacturer’s protocol.

### 4.6. ELISA 

The concentration of albumin (Alb; Bethyl Laboratories; Montgomery, TX, USA) and VEGF-A (R&D Systems; Wiesbaden-Nordenstadt, Germany) in plasma and brain homogenates and levels of interleukins 2 (IL-2), 6 (IL-6), and 10 (IL-10; BioLegend; Amsterdam, The Netherlands) in brain homogenates were quantified by enzyme-linked immunosorbent assay (ELISA) according to the manufacture’s protocols. Samples were analyzed in duplicate.

### 4.7. Immunohistochemistry 

First, 4 µm thick coronal sections of paraformaldehyde-embedded mouse brains were used at the level of the dorsal hippocampus (n = 3 per group). After heat-induced epitope retrieval, washing with Tris-buffered saline (TBS) containing 0.05% (*v*/*v*) Tween 20, and blocking with 10% normal goat serum, sections were incubated overnight at 4 °C with 1.0 µg/mL mouse monoclonal anti-mouse-OCLN (Santa Cruz Biotechnology; Heidelberg, Germany) and 0.5 µg/mL polyclonal rabbit anti-mouse-PECAM-1 antibody (Santa Cruz; Heidelberg, Germany) diluted in Dako Antibody Diluent (Agilent; Waldbronn, Germany) containing 10% normal goat serum. After washing, sections were stained with 40 µg/mL Alexa Fluor 647-conjugated goat anti-mouse immunoglobulin G (IgG [H+L]; Thermo Fisher Scientific; Schwerte, Germany) and 5 µg/mL Alexa Fluor 488-conjugated goat anti-rabbit IgG (H+L; Thermo Fisher Scientific; Schwerte, Germany;) secondary antibody for 60 min at room temperature. Cell nuclei were stained with 1.6 µg/µL 4′,6-diamidino-2-phenylindole dihydrochloride (DAPI; Roche; Grenzach, Germany) before the slides were covered in Dako Fluorescence Mounting Medium (Agilent Technologies; Waldbronn, Germany). For all incubations, a humidified chamber was used. Negative controls were created by omitting the primary antibody. All staining procedures were performed in triplicate.

Stained brain sections were acquired using either an Axio Observer Z1 with Axiocam 702 monochrome camera (Carl Zeiss; Jena, Germany; scale 0.147 µm per pixel) or a Pannoramic Midi scanner with a PCO edge 4.2 bi camera (3Dhistech; Budapest, Hungary; scale 0.163 µm per pixel). Whole-section raw images were captured and processed into TIFF files. Anatomical regions of interest (ROI) were defined according to the mouse brain atlas of Franklin and Paxinos [76]. Vessels were identified using anti-PECAM-1 immunostaining. Cerebral vessel structures were analyzed using ImageJ [77] and the image processing package FIJI [78]. Hereby, vessel area, length, and branching were quantified using vessel analysis plugins of Elfarnawany [79] and Rust et al. [80] with minor modifications. Briefly, pre-processed images were converted to 8 bit, and background staining was subtracted automatically; prior images were binarized through auto-thresholding, followed by despeckle. Next, PECAM-1^+^ area fraction was quantified and used to calculate the vascular density as the ratio of PECAM-1^+^ area to total area of the ROI, expressed as percentage. OCLN coverage was calculated as the quotient of OCLN^+^/PECAM-1^+^ double-stained vessels relative to the total PECAM-1^+^ vessel area. To determine vessel length and branching, pre-processed images were skeletonized prior to batch analysis.

### 4.8. Statistical Analysis

Data are presented as the mean and its standard error (SEM). Statistical significance was determined by one-way and two-way analysis of variance (ANOVA) with Bonferroni multiple testing correction in GraphPad software v.8.00 (San Diego, CA, USA) with a 0.05% significance level. Results with two-tailed *p* < 0.05 were considered statistically significant.

## Figures and Tables

**Figure 1 ijms-23-08693-f001:**
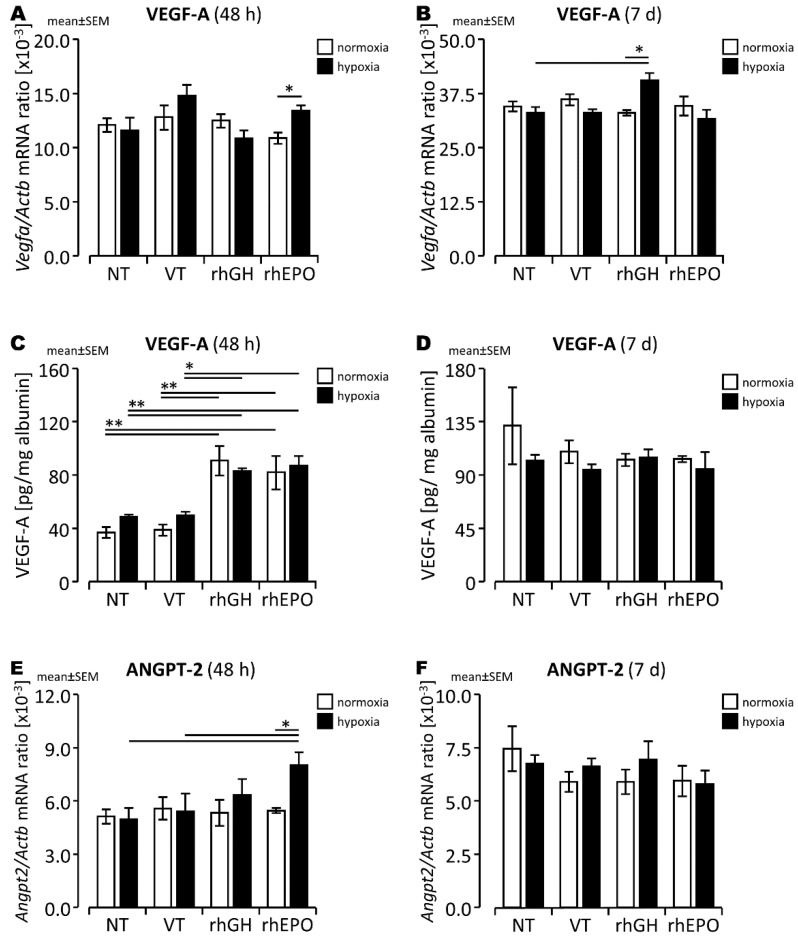
VEGF-A gene (**A**,**B**) and protein (**C**,**D**) expression and *Angpt-2* gene expression (**E**,**F**) in normoxic and hypoxic developing mouse brains after a regeneration period of 48 h (**A**,**C**) or 7 d (**B**,**D**). *, *p* < 0.05; **, *p* < 0.01.

**Figure 2 ijms-23-08693-f002:**
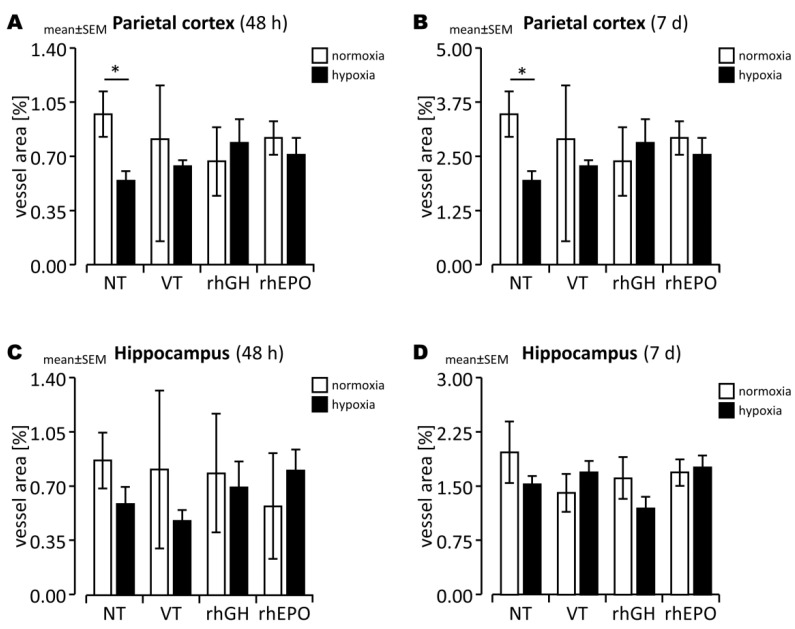
Vascular development in normoxic and hypoxic developing mouse brains with and without rhGH and rhEPO treatment. The vessel area in the parietal cortex (**A**,**B**) and hippocampus (**C**,**D**) was quantified by Pecam-1 IHC after a regeneration period of 48 h (**A**,**C**) and 7 d (**B**,**D**). Data are presented as mean ± SEM. *, *p* < 0.05.

**Figure 3 ijms-23-08693-f003:**
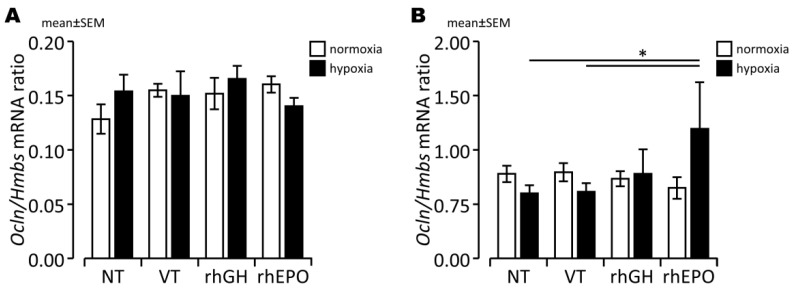
Gene expression of *Ocln* relative to PBGD in normoxic and hypoxic brains of neonatal mice with and without rhGH and rhEPO treatment after a regeneration period of 48 h (**A**) or 7 d (**B**). *, *p* < 0.05.

**Figure 4 ijms-23-08693-f004:**
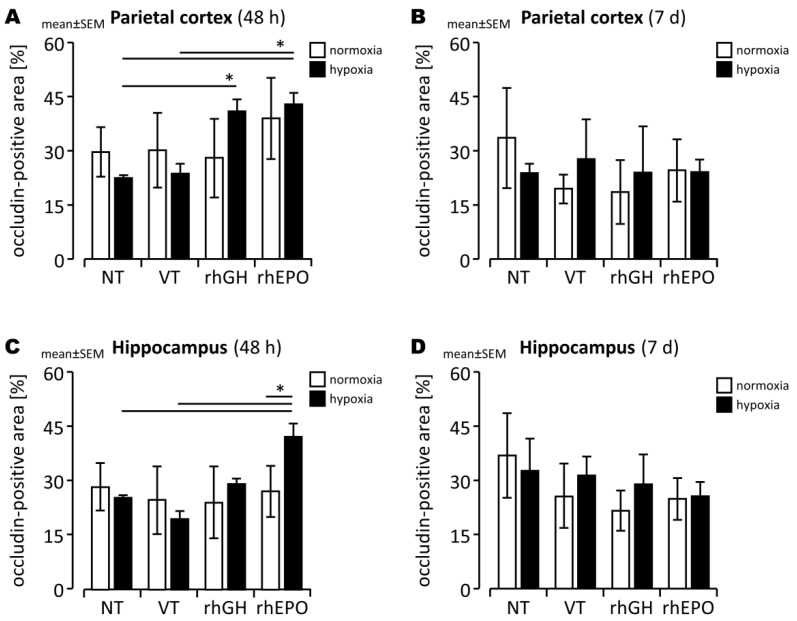
Quantification of OCLN^+^ area by immunofluorescence analysis of OCLN and PECAM-1 protein in normoxic and hypoxic developing mouse brains with and without rhGH and rhEPO treatment after a regeneration period of 48 h (**A**,**C**) and 7 d (**B**,**D**) in the parietal cortex (**A**,**B**) and hippocampus (**C**,**D**). Data are presented as mean ± SEM. *, *p* < 0.05.

**Figure 5 ijms-23-08693-f005:**
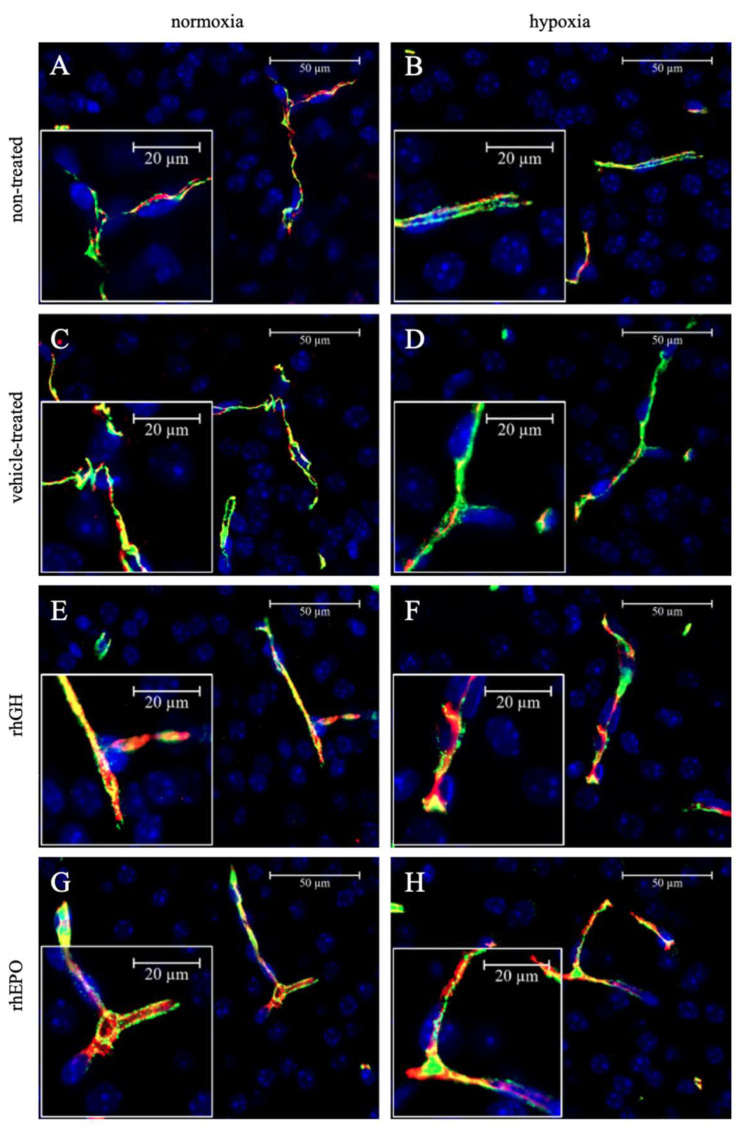
Representative photomicrographs of PECAM-1 (green) and OCLN (red) protein co-staining of vascular endothelial cells in hypoxic developing mouse brains (**B**,**D**,**F**,**H**) compared to normoxic controls (**A**,**C**,**E**,**F**) after a 48 h regeneration period in NT (A,B), VT (**C**,**D**), rhGH- (**E**,**F**), and rhEPO-treated brains (**G**,**H**) in the parietal cortex. Blue, 4′,6-diamidino-2-phenylindole (DAPI) nuclear counterstain.

**Figure 6 ijms-23-08693-f006:**
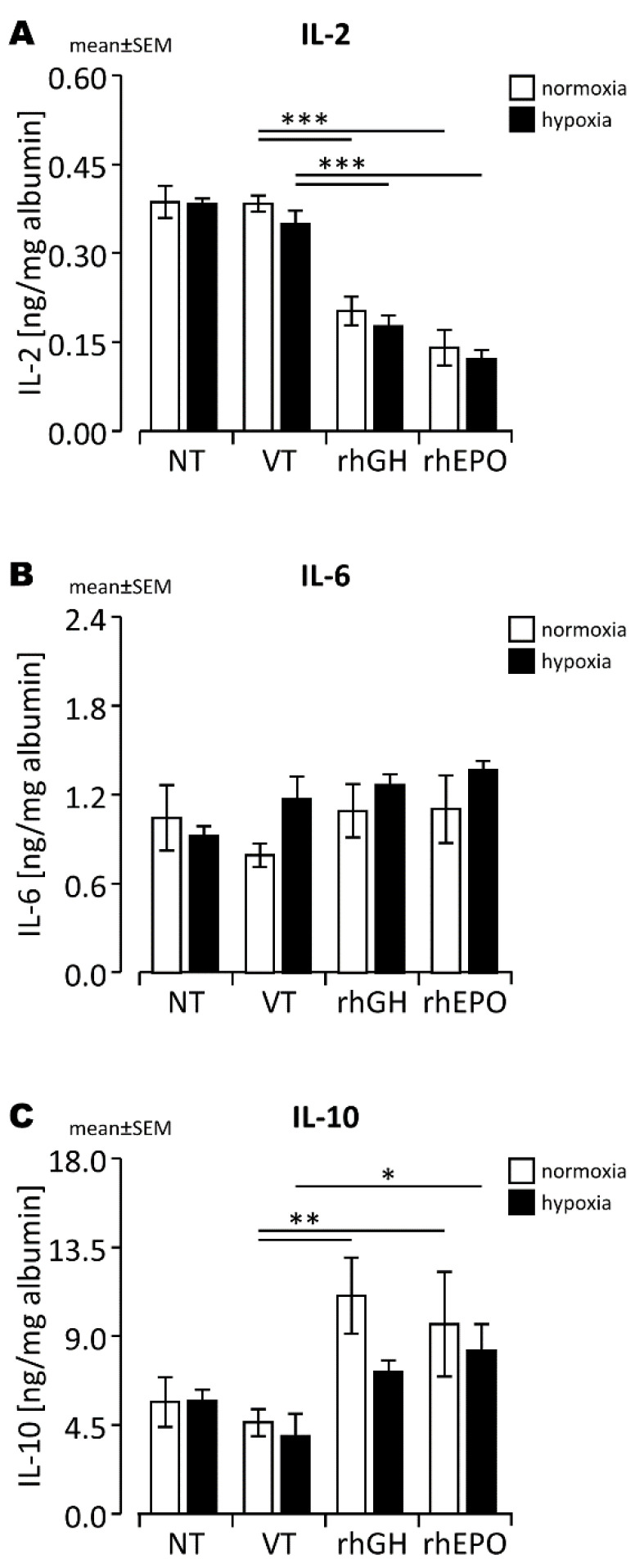
Protein levels of (**A**) IL-2, (**B**) IL-6, and (**C**) IL-10 in normoxic and hypoxic mouse brains after a 48 h regeneration period. *, *p* < 0.05; **, *p* < 0.01; ***, *p* < 0.001.

**Table 1 ijms-23-08693-t001:** qRT-PCR primers.

Target		5′-3′ Sequence
**ß-actin**	forward	5′-ATGCTCCCCGGGCTGTAT-3′
**(*Actb*)**	reverse	5′-TCACCCACATAGGAGTCCTTCTG-3′
	probe	5′(Fam)-ATCACACCCTGGTGCCTAGGGCG-(BMN-Q535)-3′
**PBGD**	forward	5′-ACAAGATTCTTGATACTGCACTCTCTAAG-3′
**(*Hmbs*)**	reverse	5′-CCTTCAGGGAGTGAACGACCA-3′
	probe	5′(Fam)-TCTAGCTCCTTGGTAAACAGGCTCTTCTCTCCA-(BMN-Q535)-3′
**VEGF-A**	forward	5′-GCACTGGACCCTGGCTTTACT-3′
**(*Vegfa*)**	reverse	5′-ACTTGATCACTTCATGGGACTTCTG-3′
	probe	5′(Fam)-CCATGCCAAGTGGTCCCAGGCTG-(BMN-Q535)-3′
**VEGFR-1**	forward	5′-TAAGACGGTTAGCACATTGGTGG-3′
**(*Flt1*)**	reverse	5′-AGTTTCAGGTCCTCTCCTTCGG-3′
	probe	5′(Fam)-TGTCACAGATGTGCCGAATGGCTTTC-3′
**VEGFR-2**	forward	5′-ACTAGGAAAACCTCTTGGCCG-3′
**(*Kdr*)**	reverse	5′-TCTTGAGTTCAGACATGAGGGCT-3′
	probe	5′(Fam)-AGATGTTGAAAGAAGGAGCAACACACAGCG-3′
**TIE-2**	forward	5′-TCAAGAGGATGAAAGAGTATGCC-3′
**(*Tek*)**	reverse	5′-TAGGTACAAATAGCCTCGGTG-3′
	probe	5′(Fam)-ATCACAGGGACTTCGCAGGAGAACTGGAG-3′
**ANGPT-1**	forward	5′-TCCACATAGGAAATGAAAAGCAGAAC-3′
**(*Angpt1*)**	reverse	5′-ACACCAACCTCCTGTTAGCAT-3′
	probe	5′(Fam)-AGGTCACACAGGGACAGCAGGCAAACAGAG-3′
**ANGPT-2**	forward	5′-CAACTACAGGATTCACCTTACAG-3′
**(*Angpt2*)**	reverse	5′-CAAACCACCAGCCTCCTG-3′
	probe	5′(Fam)-CAAAATAAGTAGCATCAGCCAACCAGGAAGTG-3′
**OCLN**	forward	5′-TCTAGATAAAGAGCTGGATGAC-3′
**(*Ocln*)**	reverse	5′-TCTTACTTTTATAATCTGCAGATCCC-3′
	probe	5′(Fam)-AGCAGCCATGTACTCTTCACTCTCCTCTCTG-3′
**Claudin-1**	forward	5′-CCCATCAATGCCAGGTATG-3′
**(*Cldn1*)**	reverse	5′-AAAGTAGGACACCTCCCAG-3′
	probe	5′(Fam)-TCTTTACTGGCTGGGCCGCTGCCTC-3′
**Claudin-5**	forward	5′-AGTTAAGGCACGGGTAGCAC-3′
**(*Cldn5*)**	reverse	5′-ATGTTGGCGAACCAGCAGAG-3′
	probe	5′(Fam)-ACGGGAGGAGCGCTTTACGCGGTGT-3′
**ZO-1**	forward	5′-TTGCCCTCACAGTACAGC-3′
**(*Tjp1*)**	reverse	5′-TGATACTGAGTTGCCTTCACC-3′
	probe	5′(Fam)-ACCTCTGTCCAGCTCTTCTCTCCACATAC-3′

## Data Availability

Data generated during and/or analyzed during the present study are available from the corresponding author on reasonable request.

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
