# Peer review of "Further Evidence of Neuroprotective Effects of Recombinant Human Erythropoietin and Growth Hormone in Hypoxic Brain Injury in Neonatal Mice"

_ijms, 2022, doi:10.3390/ijms23158693_

Round 1

Reviewer 1 Report

The manuscript by Klepper et al. investigates the neuroprotective effects of recombinant human erythropoietin (rhEPO) and growth hormone (rhGH) in hypoxia-induced brain damage using qRT-PCR, ELISA, and immunohistochemistry. Overall, the manuscript looks very interesting and the whole study seems to be acceptable. However, I have some concerns which need to be addressed.

  1. All figures and figure labels are blurred. Please, change high quality images.
  2. Please, make all graph images with the same size.
  3. Revise the figure legends. Please, add the title of the figure legends and number of mice.
  4. In figure 3 legends, statistical significance is missing.
  5. Add IHC images with quantification data of vessel area (%) (figure 2) and occludin+ vascular endothelial cells (figure 4).
  6. The authors use IL-2, IL-6 as proinflammatory cytokines. But these are expressed in a similar pattern following normoxia and hypoxia in non-treatment and vehicle treatment groups. Please discuss the possible reason behind this in the results section.
  7.  The discussion part is well written. But the authors should minimize the size of the discussion part. Please rewrite the discussion section.
  1. rhEPO and rhGH have been demonstrated significant effects in different diseases as well as under clinical trials. It is better to mention clinical trials status in the introduction part.
  2. Add full form of rhEPO in abstract (1st sentence).

Author Response

Reviewer 1

Thank you very much for evaluating our manuscript and for your helpful comments and advices.

  1. Author descripted the effects of rhEPO and rhGH in the title as neuroregenerative, while they showed the angiogenesis effects and protective effects on hypoxia-induced BBB disruption in this manuscript. A title should indicate the conclusion more clearly and straightly.

Reply: Thank you very much for this constructive comment. We changed the title of the manuscript according to the recommendation of reviewer #1.

  1. In the material and methods, authors said the cerebral vessel structures were identified and quantified using ImageJ (Schneider et al., 2012), Please conform the reference and descript the methods in detail.

Reply: We apologize for the short description of how we identified and quantify the vascular area fraction, vessel length and branching. For better understanding, we extended this section. To ensure an unbiased analysis, images were recorded with identical settings and batch-processed using the open-source image software ImageJ in combination with the image processing package Fiji. Vessel characteristics were analyzed using plugins and macros written by experts in the field of vessel analysis; respective references were included.

  1. In figure 2, vessel area was quantified by Pecam-1 IHC as authors claimed. Please show the original IHC staining results and detailed description of the quantify methods.

Reply: We agree with this reviewer´s comment and included representative images of PECAM-1+ vessel staining in the supplemental material (Supplementary Figure 3). A detailed description was added. As mentioned above, image pre-processing and analysis was performed in an automatic manner using ImageJ and Fiji. Briefly, vessels were identified using anti-PECAM-1 immunostaining. Vessel area was calculated as the ratio of vasculature Pecam-1+ area to total area of the ROI (see method section).

  1. In figure 4, please add the original immunofluorescence staining results and detailed description of cell counting.

Reply: Thank you very much for this comment. Unfortunately we made a mistake in the labeling of the y-axis which we want to correct and for which we want to apologize. Instead of “occludin-positive cells [%]” it has to be “occludin-positive area [%]” mean. We mentioned in method section that Ocln coverage was calculated as the quotient of Ocln+/Pecam-1+ double-stained vessels relative to the total Pecam-1+ vessel area. Representative images of the original immunofluorescence staining are presented in Figure 5 of the revised manuscript.

  1. Figure 1,2,3,5,6 were in low resolution, please change them to high resolution.

Reply: We apologize for the insufficient quality of the figures. It seems that the resolution was significantly reduced during the uploading process. We thank you for the hint and reformatted them to a less susceptible image format.

Reviewer 2 Report

In this manuscript, researchers reported that application of rhEPO and rhGH increased the expression of Vegf-a and Angpt-2, indicated the regenerative vascular effects and the protective effects on hypoxia-induced BBB disruption. The angiogenesis effects of EPO and GH had been reported in many studies, while in this manuscript, the authors showed the direct evidence about the angiogenesis effects in the mice model of neonatal hypoxia, inspected the effect of EPO and HG. Here are the comments

1.  Author descripted the effects of rhEPO and rhGH in the title as neuroregenerative, while they showed the angiogenesis effects and protective effects on hypoxia-induced BBB disruption in this manuscript.  A   title should indicate the conclusion more clearly and straightly.

2.    In the material and methods, authors said the cerebral vessel structures were identified and quantified using ImageJ (Schneider et al., 2012), Please conform the reference and descript the methods in detail.

3.  In figure 2, vessel area was quantified by Pecam-1 IHC as authors claimed. Please show the original IHC staining results and detailed description of the quantify methods.

4.  In figure 4, please add the original immunofluorescence staining results and detailed description of cell counting.

5.  Figure 1,2,3,5,6 were in low resolution, please change them to high resolution.

Author Response

Reviewer 2

We were pleased to note that you consider our work as “original and very interesting”. We thank you very much for the evaluation of our manuscript and helpful comments. 

I suggest to the authors, if it is possible, to improve the paper insert the localization of IL2, IL 6, and IL10 in the brain tissue. The authors correctly performed an ELISA test on plasma and brain homogenates tissue. Still, I think that it could be interesting to know the localization of cytokines using immunohistochemistry or immunofluorescence. If the authors don't have additional brain samples I suggest improving the discussion with the analysis literature about cytokines brain expression and hypoxia.

Reply: Thank you very much for these helpful comments. Due to national and European laws on the protection of animals, the number of animals used for this study was strictly limited. Thus, we have no more brain tissue left to perform the requested immunohistochemistry or immunofluorescence analyses. To address the neuroinflammatory response we analyzed IL-2, IL-6, and IL-10 expression in whole brain lysates. We agree that cell type-specific analysis could give deeper insights in regulatory mechanisms, however, this was beyond the scope of our present study. As suggested from (limited) literature, microglia play a central role in hypoxia-induced inflammatory neurodegeneration in the developing brain. According to this reviewer’s helpful comment we reworded the discussion section.

Reviewer 3 Report

The paper entitled "Further evidence of neuroregenerative effects of recombinant human erythropoietin and growth hormone in hypoxic brain injury in neonatal mice", is original and very interesting.

I suggest to the authors, if it is possible, to improve the paper insert the localization of IL2, IL 6, and IL10 in the brain tissue. The authors correctly performed an ELISA test on plasma and brain homogenates tissue. Still, I think that it could be interesting to know the localization of cytokines using immunohistochemistry or immunofluorescence. If the authors don't have additional brain samples I suggest improving the discussion with the analysis literature about cytokines brain expression and hypoxia.

Author Response

(The authors gave the same response as above.)
